# An In Vitro Pilot Fermentation Study on the Impact of *Chlorella pyrenoidosa* on Gut Microbiome Composition and Metabolites in Healthy and Coeliac Subjects

**DOI:** 10.3390/molecules26082330

**Published:** 2021-04-16

**Authors:** Carmen van der Linde, Monica Barone, Silvia Turroni, Patrizia Brigidi, Enver Keleszade, Jonathan R. Swann, Adele Costabile

**Affiliations:** 1Department of Life Sciences, University of Roehampton, London SW15 4JD, UK; carmen.linde8@gmail.com (C.v.d.L.); keleszae@roehampton.ac.uk (E.K.); 2Unit of Microbial Ecology of Health, Department of Pharmacy and Biotechnology, University of Bologna, 40126 Bologna, Italy; monica.barone@unibo.it (M.B.); silvia.turroni@unibo.it (S.T.); 3Department of Medical and Surgical Science, University of Bologna, 40126 Bologna, Italy; patrizia.brigidi@unibo.it; 4School of Human Development and Health, Faculty of Medicine, University of Southampton, Southampton SO16 6YD, UK; j.swann@soton.ac.uk; 5Division of Systems Medicine, Department of Metabolism, Digestion and Reproduction, Imperial College London, London SW7 2AZ, UK

**Keywords:** *C. pyrenoidosa*, in vitro gut model, gut microbiome, metabolism

## Abstract

The response of a coeliac and a healthy gut microbiota to the green algae *Chlorella pyrenoidosa* was evaluated using an in vitro continuous, pH controlled, gut model system, which simulated the human colon. The effect of *C. pyrenoidosa* on the microbial structure was determined by 16S rRNA gene sequencing and inferred metagenomics, whereas the metabolic activitywas determined by^1^H-nuclear magnetic resonancespectroscopic analysis. The addition of *C. pyrenoidosa* significantly increased the abundance of the genera Prevotella, *Ruminococcus* and *Faecalibacterium* in the healthy donor, while an increase in *Faecalibacterium*, *Bifidobacterium* and *Megasphaera* and a decrease in Enterobacteriaceae were observed in the coeliac donor. *C. pyrenoidosa* also altered several microbial pathways including those involved in short-chain fatty acid (SCFA) production. At the metabolic level, a significant increase from baseline was seen in butyrate and propionate (*p* < 0.0001) in the healthy donor, especially in vessels 2 and 3. While acetate was significantly higher in the healthy donor at baseline in vessel 3 (*p* < 0.001) compared to the coeliac donor, this was markedly decreased after in vitro fermentation with *C. pyrenoidosa*. This is the first in vitro fermentation study of *C. pyrenoidosa* and human gut microbiota, however, further in vivo studies are needed to prove its efficacy.

## 1. Introduction

Humans only have the ability to degrade two types of glycosidic linkages found in glycans and therefore cannot degrade most of the complex carbohydrates present in the diet [1]. Collectively known as dietary fibre or microbiota-accessible carbohydrates (MACs), these indigestible plant polysaccharides are degraded and fermented by the gut microbiota as its main energy source [2]. Analyses of microbial genes have shown a variety of carbohydrate-active enzymes (CAZymes) enabling the breakdown of diverse polysaccharide structures in the gut [3,4]. The decline in MAC dietary intake in the urban Western population (lower than the recommended 30 g of fibre/day) has been linked to decreased bacterial diversity, altered immunity and disease progression [5]. MACs also have the ability to modulate the gut microbiota towards a more health-promoting diverse profile. Recent studies have demonstrated the impact of low dietary MACs in mice, showing a reduction in bacterial diversity and a thinning of the mucus layer in the gut [6]. Conversely, Sonnenburg et al. (2016) showed that switching from a low- to a high-MAC diet in mice could largely restore microbiota diversity [7].

In a gut deprived of fermentable fibre, the microbiota can degrade the mucus layer, which is rich in glycoproteins, as an energy source leading to closer contact between the bacteria and the gut epithelial layer. This can disrupt the normal gut barrier function and increase the susceptibility to invasion by pathogenic microorganisms [6]. 

Coeliac disease (CD) mainly involves small intestinal inflammation with villous atrophy and is characterised by an altered microbiome layout [8,9,10,11,12]. Individuals with CD tend to have larger numbers of Gram-negative bacteria (i.e., Bacteroidetes and Proteobacteria) and a lower abundance of Gram-positive bacteria (i.e., Actinobacteria and Firmicutes) in their gut [13,14], as well as a shift towards facultative anaerobes [15]. However, the individual microbes involved in CD development and progression and the underlying mechanisms remain elusive [12].

To date, a lifelong strict gluten-free diet (GFD) is the only treatment option for people diagnosed with CD [16]. A GFD and a reduction in polysaccharides have been shown to alter the CD microbiota, with a reduction in *Bifidobacterium* sp., *Faecalibacterium prausnitzii* and *Lactobacillus* sp. and an increase in *Enterobacteriaceae* [9,10,11,12].

Along with microbiome alteration, unique changes in the faecal profiles of short-chain fatty acids (SCFAs) are observed in CD individuals undergoing a GFD [10,17]. SCFAs are important end products of microbial fermentation, which act as an energy source for colonocytes and provide a suitable colonic environment for optimal functioning [18]. However, changes in SCFAs production due to alterations in microbiota composition and function can alter the energy balance and contribute to the onset of metabolic diseases [19,20].

*Chlorella* sp. is a unicellular green algae that has become popular as a food supplement, especially in Asia, due to its high nutritional value and content of amino acids, vitamins, polyunsaturated fatty acids, chlorophyll, as well as bioactive compounds [21]. It has also gained interest as a functional food [22,23,24,25] with hypoglycaemic and cholesterol lowering effects corresponding to microbiota changes in animal studies [26,27].

To date, data have shown that the content of natural compounds in *Chlorella* differs greatly between culture conditions and *Chlorella* species [28,29]. The cell wall of *C. pyrenoidosa* consists of 9.2% α-cellulose, which is composed of glucose, galactose, arabinose, mannose, xylose and rhamnose. Hemicellulose constitutes 15.4% of the cell wall, containing mainly galactose, while glucosamine contributes 31% to the cell wall [30]. Algal cell walls contain large amounts of polysaccharides that can act as dietary fibre and have the potential to be fermented by colonic bacteria [31]. Particularly, a polysaccharide isolated from *C. pyrenoidosa* has been found to contain several types of glycosidic bonds along with seven different monosaccharides [27]. *C. pyrenoidosa* also contains several polyphenols like flavonoids, lutein, epigallocatechin gallate and catechin [32].

*C. pyrenoidosa* has been shown to prolong the hypoglycaemic effects of exogenous insulin in both normal and diabetic mice [33]. An improvement in insulin sensitivity has also been observed in diabetic rats, with a dose-dependent decrease in blood glucose levels in rats fed a high-fructose diet [26]. Eleven weeks of *Chlorella* supplementation also significantly reduced glycated haemoglobin (HbA1c) and total serum cholesterol in diabetic rats [34]. A recent study by Wan et al. (2019) suggests that the hypoglycaemic effects of *C. pyrenoidosa* are correlated with changes in the gut microbiota [27]. An eight-week intervention with *C. pyrenoidosa* extract reduced serum cholesterol and triglycerides in rats fed a high-fat diet, also indicating a protective effect by decreasing liver fat accumulation along with an increased abundance of *Alistipes*, *Prevotella* and *Ruminococcus* [27]. Interestingly, after *Chlorella* supplementation, a negative correlation has been observed between fasting blood glucose (FBG) and *Ruminococcus, Parasutterella* and *Alloprevotella*, together with a positive correlation with *Lactobacillus* [27]. Likewise, a pivotal study of a large European female cohort showed that *Lactobacillus* sp. are significantly positively correlated with FBG and HbA1c, while *Clostridium* sp. had a negative correlation with FBG, HbA1c and plasma triglycerides [35].

Here, we used an in vitro continuous, pH controlled gut model system that reflected the environmental conditions of the three regions of the human colon (vessel 1, proximal colon; vessel 2, transverse colon; vessel 3, distal colon)to assess whether *C. pyrenoidosa* has the potential to modulate the coeliac gut microbiome towards a healthier configuration, thereby affecting the production of SCFAs. This study aims therefore to examine the outcome of changes in the microbiota composition (and inferred function) and metabolite profile during an in vitro fermentation with *C. pyrenoidosa* between coeliac and healthy human donors. 

## 2. Materials and Methods

### 2.1. C. pyrenoidosa 

*C. pyrenoidosa* (Sun Chlorella ‘A’ Powder) individual powder sachets (3 g) pulverized by DYNO-Mill and supplied by Sun Chlorella Corporation (Kyoto, Japan). This remarkable technology liberates the nutrients in *Chlorella*, whilst breaking up the in-digestible *Chlorella* cell wall without the need for excessive heat or chemicals. The nutritional composition is reported in Appendix A.

### 2.2. Faecal Sample Preparation 

Faecal samples were obtained from a 30-year-old female with normal gluten-containing diet (H) and a 30-year-old female with Coeliac disease who had been following a GFD for more than 1 year. Both participants had not taken antibiotics and/or prebiotics or probiotic supplements for at least 3 months before sampling. An aliquot of 20 g of faecal samples from healthy (H) and coeliac (C) volunteer was diluted in 100 mL anaerobic PBS (0.1 mol/L phosphate buffer solution, pH 7.4, *w/w*) and homogenised (Stomacher 400; Seward, West Sussex, UK) for 2 min at 240 paddle beats per minute.

The University of Roehampton Research Ethics Committee (LSC 18-241) approved this study in accordance with the Declaration of Helsinki. Sample size was in accordance with previous studies [36].

### 2.3. Three-Stage Continuous Culture Gut Model System

A small-scale version of the validated three-stage continuous system was used in this study, consistent with previous studies [37] with vessels (V) representing the proximal (V1, 80 mL, pH = 5.5), transverse (V2, 100 mL, pH = 6.2) and distal colon (V3, 120 mL, pH = 6.8). A schematic chart of the validated three-stage continuous system is reported in Appendix A. Each vessel was magnetically stirred and continually sparged with oxygen free nitrogen gas. Temperature (37 °C) was maintained by a water-cooling system and culture pH was controlled automatically through the addition of 1 N NaOH or HCl [37].

All vessels were filled with the faecal homogenate as previously described and a complex colonic growth medium (80%). The growth medium contained the following ingredients: starch, 5 g/L; mucin, 4 g/L; casein, 3 g/L; peptone water, 5 g/L; tryptone water, 5 g/L; bile salts, 0.4 g/L; yeast exact, 4.5 g/L; FeSO_4_, 0.005 g/L; NaCl, 4.5 g/L; KCl, 4.5 g/L; KH_2_PO_4_, 0.5 g/L; MgSO_4_ × 7H_2_O, 1.25 g/L; CaCl_2_ × 6H_2_O, 0.15 g/L; NaHCO_3_, 1.5 g/L; Tween 80, 1 mL; hemin, 0.05 g/L; and cysteine HCl, 0.8 g/L. 

Following inoculation, the colonic model was run as a batch culture for 24 h to stabilize the bacterial populations prior to the initiation of medium flow. 

After 24 h, the medium flow was initiated and the system was run for eight full volume turnovers at 15, 16 and 17 days to allow for steady state to be achieved (SS1). The duration of each turnover was 48 h thus this first step was done for 384 h with the flow rate of the system set at 6.25 mL/h. Thereafter, *C. pyrenoidosa (Sun Chlorella* ‘A’ Powder) was administered into V1 in a powder form3 g dose at 10.00 am each day until the second steady state (SS2) had been reached at 33, 34, and 35 days. Each steady state was confirmed by stabilization of SCFAs profiles over 3 consecutive days. SS2 was achieved after 384 h; the whole experiment took 792 h. A sample of 4.5 mL was taken through the sample port of each vessel (V1, V2 and V3) subsequent to the first (SS1) and second full turnover (SS2) of medium through the model. Two aliquots of 2.25 mL each were centrifuged at 13,000 rpm, the supernatant was stored at −20 °C for short chain fatty acid profile whereas the pellets were stored at −80 °C for 16S rRNA gene-based next-generation sequencing analysis.

#### ^1^H-Nuclear Magnetic Resonance Spectroscopic (^1^H-NMR) Metabolomic Profile of Supernatants from Fermentation

Sample analysis was performed according to the procedure outlined by Grimaldi et al. (2018) [38]. Spectral data was imported into Matlab (Mathworks, version 2018a) using in-house scripts. Spectral regions containing redundant peaks (H_2_O, trimethylsilylpropionate (internal standard]) were excised from the dataset and the profiles were normalized using a probabilistic quotient approach. Principal components analysis (PCA) was performed on the ^1^H-NMR spectral data in Matlab using in-house scripts. Based on the PCA findings, the peak integrals (representing relative abundance) for acetate (δ 1.92, singlet), butyrate (δ 0.90, triplet) and propionate (δ 1.06, triplet) were calculated. Samples were collected in duplicate before intervention for SS1 and SS2 as previously described from each vessel (V1, V2, V3) from both gut models with Coeliac (C) and healthy (H) donors. 

### 2.4. Microbial DNA Extraction

Total microbial DNA was extracted from 250 mg of fermentation samples using the QIAamp DNA Stool Mini Kit (QIAGEN, Hilden, Germany) according to the manufacturer’s instructions. DNA concentration and quality were evaluated using the NanoDrop ND-1000 spectrophotometer (NanoDrop Technologies, Wilmington, DE, USA).

### 2.5. 16S rRNA Gene-Based Next-Generation Sequencing and Bioinformatics

DNA samples were analysed by 16S rRNA gene sequencing according to the method described by Corona et al. (2019) [39]. In brief, for each sample, the hypervariable V3−V4 regions of the 16S rRNA gene were PCR-amplified using the S-D-Bact-0341-b-S-17/S-D-Bact-0785-a-A-21 primers with Illumina overhang adapter sequences [40]. PCR reactions were performed in a final volume of 25μL, containing 12.5 ng of genomic DNA, 200 nM of each primer, and 2X KAPA HiFi HotStart ReadyMix (Kapa Biosystems, Roche, Basel, Switzerland). The following gradient was used: 3 min at 95 °C for initial denaturation, 25 cycles of denaturation at 95 °C for 30 s, annealing at 55 °C for 30 s and extension at 72 °C for 30 s, followed by a final extension step at 72 °C for 5 min. PCR products were purified by using a magnetic bead-based system (Agencourt AMPure XP; Beckman Coulter, Brea, CA, USA), indexed by limited-cycle PCR using Nextera technology, and further purified as described above. Final libraries were pooled at equimolar concentration (4 nM), denatured with 0.2 N NaOH, and diluted to 6 pM before loading onto the MiSeq flow cell. Sequencing was performed on an Illumina MiSeq platform with the 2 × 250 bp paired-end protocol, according to the manufacturer’s instructions (Illumina, San Diego, CA, USA). All sequencing reads were deposited in the National Center for Biotechnology Information Sequence Read Archive (Bioproject: PRJNA685154).

Raw reads were processed using a pipeline combining PANDAseq [41] and QIIME 2 [42]. High-quality reads were filtered and clustered into amplicon sequence variants (ASVs) using DADA2 [43]. Taxonomy was assigned using the VSEARCH classifier [44] against Greengenes database as a reference (release May 2013). Alpha diversity was measured using the number of observed ASVs and the Faith’s Phylogenetic Diversity (PD whole tree) index. Beta diversity was computed based on weighted and unweighted UniFrac distances and visualized on a Principal Coordinates Analysis (PCoA) plot. Metagenome prediction of Greengenes-picked ASVs was performed with PICRUSt2) [45], using MetaCyc [46] as reference for pathway annotation.

### 2.6. Data Analysis 

Statistical analyses were performed using GraphPad Prism 8 and RStudio. Matlab was used to plot the PCA of the metabolite dataset.

Data from ^1^H-NMR were analysed employing a paired t-test, comparing before (SS1) and after the treatment (SS2) with *C. pyrenoidosa* for each SCFA (acetate, butyrate, propionate) for both Coeliac and healthy gut models. The statistical significance level was set at *p* < 0.1 due to the small sample size. As for the microbiota data, statistics were performed using R Studio 1.0.44 on R software version 3.3.2 (https://www.r-project.org (accessed on 13 February 2021)) [47] implemented with the packages stats and vegan (https://cran.r-project.org/web/packages/vegan/vegan.pdf (accessed on 13 February 2021)) [48].

The significance of data separation in the PCoA plot was tested by a permutation test with ANOSIM and pseudo-*F* ratio statistics using the function adonis in vegan. Bar plots were built using the R packages made4 [47] and vegan. Sample clustering was performed according to the pathway-level abundance profiles, using Kendall’s correlation coefficients as metric and Ward-linkage method. Linear discriminant analysis effect size (LEfSe) algorithm with LDA score threshold of 2 (on a log10 scale) was applied to identify the MetaCyc pathways affected by the addition of *C. pyrenoidosa* [49]. Nonparametric tests (Kruskal–Wallis test or Wilcoxon test, paired or unpaired as needed) were achieved using the stats package. A *p* ≤ 0.05 was considered statistically significant; a P between 0.05 and 0.1 was considered a tendency.

## 3. Results

### 3.1. Impact of C. pyrenoidosa on Faecal-Derived Microbial Communities from Healthy and Coeliac Donors

The faecal-derived microbial communities from a healthy donor (H) and a Coeliac donor (C) were profiled at baseline (steady state 1, SS1) and after the addition of *C. pyrenoidosa*, upon reaching steady state 2 (SS2). To ensure that effects were due solely to the treatment with *C. pyrenoidosa* and not to the adaptation of microbes to the in vitro environment, steady-state 1 and steady-state 2 (conditions in terms of microbial community composition and metabolic activity) were needed to be established prior to the actual start of the experiment. The 16S rRNA gene-based next-generation sequencing of all fermentation samples yielded a total of 868,020 high-quality reads, with an average of 36,168 ± 8485 sequences per sample, binned into 2289 amplicon sequence variants (ASVs).

No significant differences were observed in alpha diversity across the entire dataset, regardless of the origin of the faecal sample (C vs. H), simulated intestinal region (vessel 1 vs. 2 vs. 3) and time point (baseline vs. steady state) (*p* > 0.05, Kruskal−Wallis test) (Figure 1A). However, a trend towards lower diversity was found after addition of *C. pyrenoidosa* for both donors. This trend was more marked in vessels 1 and 3 than in vessel 2.

Principal coordinate analysis (PCoA) based on weighted and unweighted UniFrac distance matrices showed separation between the faecal-derived microbial communities from C and H donors, regardless of the time point and the fermentation vessel (*p* = 0.002 and 1 × 10^−4^, respectively; permutation test with pseudo-*F* ratio) (Figure 1B,C). Furthermore, the analysis of temporal dynamics revealed clearly distinct trends according to the donor, suggesting a differential impact of *C. pyrenoidosa* on the microbiota structures, very likely to be related to the initial configuration. These trends were generally consistent among vessels, except for vessel 1 inoculated with C faeces, for which no substantial variations were observed after *C. pyrenoidosa* addition according to the weighted UniFrac distances, and vessel 3 inoculated with H faeces, for which a contrasting trend was observed in the weighted UniFrac PCoA space compared to the other two vessels.

At baseline, several taxonomic differences were observed between the microbiota of the C versus H donor (Appendix A). In particular, compared to donor H, the faecal-derived microbial ecosystem of the C donor showed a greater relative abundance of *Bacteroidaceae* (i.e., *Bacteroides*), *Enterobacteriaceae* and *Clostridiaceae* families (*p* ≤ 0.2, Wilcoxon test). In contrast, *Bifidobacteriaceae* (i.e., *Bifidobacterium* sp.), *Alcaligenaceae* and *Veillonellaceae* were more represented in donor H than in C (*p* ≤ 0.2).

When looking for differences in microbiota composition between baseline and steady state following *C. pyrenoidosa* supplementation, we observed both common and unique microbial signatures of response (Figure 2 and Appendix A). Among the features shared between C and H donors, it is worth noting that the families *Ruminococcaceae* and *Xanthomonadaceae* substantially increased after *C. pyrenoidosa* addition in all vessels, while a general decrease was found for *Bacteroidaceae* and *Erysipelotrichaceae* (*p* ≤ 0.1, Kruskal−Wallis test) (Appendix A). On the other hand, some taxa showed distinct trends based on the origin of the faecal sample. Specifically, *Prevotellaceae* increased only in H samples, while the relative abundance of *Enterobacteriaceae* decreased in C while it increased in H samples (*p* ≤ 0.06). Although in the absence of statistical significance, the relative abundance of *Clostridiaceae* decreased over time, especially in C samples, while that of *Alcaligenaceae* and *Veillonellaceae* tended to increase in C samples and decrease in H samples. A trend towards increased proportions in C samples was also observed for *Bifidobacteriaceae*, especially in vessels 2 and 3 (*p* = 0.09). Consistent data were obtained at the genus level for *Bifidobacterium* sp., *Bacteroides*, *Prevotella*, the *Ruminococcaceae* members *Faecalibacterium* and *Oscillospira*, and *Klebsiella* (*Enterobacteriaceae* family) (*p* ≤ 0.09) (Figure 2). In addition, significant differences were observed for *Lachnospira* and *Akkermansia*, whose proportions decreased overall in all vessels regardless of the faecal sample origin, *Ruminococcus* sp., whose amounts increased only in H samples, and *Megasphaera* (*Veillonellaceae* family), the amounts of which instead increased only in C samples (*p* ≤ 0.04).

To gain insight into the functional variations of the C and H microbiome in the presence of *Chlorella pyrenoidosa*, we expanded the microbiota data analysis to better elucidate the functional profiles of the microbiota. The respective metagenomes were inferred from the phylogenetic profiles using PICRUSt2, we used the LEfSe algorithm in order to identify the MetaCyc pathways affected by the addition of the *C. pyrenoidosa* extract. According to this analysis, the faecal-derived microbial communities from H and C donors showed distinct functional profiles that were differentially modulated by *C. pyrenoidosa*. Samples were then clustered according to the abundance profile of the 238 most abundant identified pathways (Figure 3A), showing an overall tendency towards segregation by origin (C vs. H). According to a LEfSe analysis (Figure 3B), C samples showed a distinct functional profile at baseline compared to H samples, characterized by an enrichment in pathways involved in enterobactin biosynthesis, fatty acid beta-oxidation I and the superpathway of glycolysis, pyruvate dehydrogenase, TCA, and glyoxylate bypass. In the contrast, evolutionarily conserved biochemical pathways, such as UMP biosynthesis I, pyrimidine nucleobases salvage I, UTP and CTP de novo biosynthesis were underrepresented in C versus H samples. Among the common effects induced by coincubation with *C. pyrenoidosa*, we found an increased functional potential involved in guanosine and adenosine nucleotide metabolism (degradation and de novo biosynthesis), together with a reduced contribution of the superpathway of L-tryptophan biosynthesis. In contrast, an increased potential for the biosynthesis of thiazoles I (by facultative anaerobic bacteria), and a greater contribution of the superpathway of thiamine diphosphate biosynthesis II were exclusively observed in C samples. With specific regard to SCFAs production, in C donor, *C. pyrenoidosa* addition led to the increase of the pathway acetyl-CoA fermentation to butanoate II, and to the depletion of hexitol fermentation to lactate, formate, ethanol and acetate.

Collectively, our results confirm the potential of *C. pyrenoidosa* to modulate the composition and functionality of the human intestinal microbiota with a differential impact closely related to the initial microbial configuration.

### 3.2. Differential Impact of C. pyrenoidosa on SCFAs Production in Healthy and Coeliac Gut Model Samples

Baseline relative abundances of faecal SCFAs in the Coeliac and healthy donors are reported in Figure 4 for all three vessels. In the healthy donor (Figure 4H), a significant increase in butyrate and propionate relative concentration was seen in vessels 2 and 3 from baseline after the daily administration of *C. pyrenoidosa* extract (*p* < 0.001). In the Coeliac donor, a significant decrease in acetate was seen in vessel 3 after the addition of *C. pyrenoidosa* (*p* < 0.001), whereas a slight increase in acetate was seen in vessel 1. Furthermore, propionate was significantly increased in vessel 3 (*p* < 0.001) from SS1 to SS2 and butyrate was increased in vessel 3 between these time-points.

## 4. Discussion

The human intestine is more densely populated with microorganisms than any other organ, and is a site where they exert strong influences on human biology. With this in mind, the entire system of the gut microbiota of each individual can be pictured as a ’microbial complex ecosystem’ within the intestine, where complicated interaction networks are established, which contributes to diverse mammalian processes including protective functions against pathogens. Here, we are presenting the first exploratory pilot study investigating the effect of *C. pyrenoidosa* in two in vitro continuous culture gut models, where one was inoculated with a healthy faecal sample and the other with a faecal sample from a Coeliac donor following a gluten-free diet (GFD).

The gut model systems represent an innovative technological platform that permits the modeling of the dynamic nature of the gastrointestinal tract, allowing the adaptation of various parameters, including dilution rate, retention time, pH and temperature, to meet and maintain optimal growth conditions [50,51,52].

This model has the potential to aid in providing an understanding of how the human gut microbiota is affected by *C. pyrenoidosa*.

The findings of this proof-of-concept study, including recognition of the limitations of the model, the use of a limited number of faecal samples suggest its future development as a tool for investigating the impact of nutrition and dietary supplementation on the gut microbiota.

Models, by definition, are not exact representations of reality. We acknowledge that a model of the human microbiota will never be an exact replica of the real bacterial community within the human large intestine. The model will, however, replicate aspects of the microbial community to allow interrogation and experimental manipulation in a more convenient and less invasive way.

The resemblance to reality is the major feature for a successful simulation and could be based both on the complexity of the mock microbiota or the environmental parameters of the reactors. Although these systems do not incorporate host factors, such as intestinal secretions, immunology, or absorption, it does offer an inexpensive and reliable tool for modeling the microbial ecology and activity of the colon [37,53,54,55]. The system was run until the microbial communities reached a steady state (this typically took 8 turnovers or 16 days) that was confirmed after obtaining consistent SCFA measurements on three consecutive days [33]. Thereafter, a specific test substrate may be tested until a second steady state (SS2) is reached (again based on stable SCFA measurements), which is then compared to the original steady state or baseline. Therefore, the results of intervention at SS2 can be directly compared to that of SS1; in this way the use of a single system can still provide relevant results [56,57,58]. Moreover, to reduce the potential antagonistic effects from a pool of several donors, a faecal single inoculum from one donor can be justified as the microbiota of one donor might sense and defense the alien microbiota of a different donor.

However, the robustness of each model depends on the certainty that any observed effects on microbial composition and metabolic activity are due solely to the applied experimental treatment and not to gut microbial adaptation to the simulated gut environment. This presumes that a compositional stability of the in vitro gut ecosystem is in place prior to applying any experimental treatment and that this pseudo-steady state can reproducibly generate identical microbial communities which are the basis for comparison with other treatments [59,60,61].

There is an intricate relationship between diet, the gut microbiota and the metabolites it produces. Coeliac disease and the consequent GFD being followed have been shown to increase the risk of metabolic syndrome [62]. *C. pyrenoidosa* has demonstrated hypoglycaemic properties associated with microbiota changes in mice [27]. However, no studies to date have investigated *C. pyrenoidosa* and its potential effect on the human gut microbiota. This is the first pilot study to provide potentially new insight into the effects of *C. pyrenoidosa*, a unicellular green algae being shown much nutritional interest due to the presence of many substances that could be beneficial for human health.

A low relative abundance of *Bifidobacterium* sp. and higher levels of Proteobacteria, especially Enterobacteriaceae, have been found in subjects with Coeliac disease [63,64] and in healthy adults following a GFD [9]. The aberrant microbial profile observed in Coeliac disease has been associated with alterations in faecal metabolite production with higher concentrations of SCFAs noted in the faeces compared to healthy individuals [10,17]. The results of our study are promising and are suggesting distinct differences in the faecal microbial communities and production of SCFAs between the healthy and Coeliac donor at the baseline. In particular, the Coeliac donor, following a GFD, had significantly higher amounts of acetate, butyrate and propionate, and showed a microbial profile with the changes listed above in addition to increased proportions of Bacteroidaceae and Clostridiaceae, while reduced proportions of Alcaligenaceae and Veillonellaceae.

The addition of *C. pyrenoidosa* led to several changes in the intestinal microbial ecosystem, some common and others unique to the donor state. It is noteworthy that *C. pyrenoidosa* supplementation increased the abundance of Ruminococcaceae (especially the butyrate producers *Faecalibacterium* sp. and Oscillospira) and decreased the proportion of the mucus degrader Akkermansia [65], regardless of the donor and vessel. In addition, C. *pyrenoidosa* modified the microbial profiles of the Coeliac vessels towards a closer resemblance of the healthy donor, especially for Bifidobacteriaceae, Enterobacteriaceae and Bacteroidaceae. In a previous study, a significant positive correlation between Proteobacteria and the SCFAs, acetate and propionate was observed in children with Coeliac disease [62]. The decrease in Enterobacteriaceae found in the Coeliac gut model after the addition of *C. pyrenoidosa*, coupled with the significant decrease in acetate (and propionate in the proximal vessel), therefore suggests that Enterobacteriaceae could be responsible for elevated SCFA profiles seen in Coeliac disease. Similarly, the Bacteroidaceae family includes well-known producers of acetate and propionate, and its decrease in Coeliac samples after *C. pyrenoidosa* supplementation suggests its additional contribution to the altered SCFA profiles. Finally, we cannot exclude that acetate also decreases due to cross-feeding between Bacteroidetes and Firmicutes (especially including Megasphaera, whose increase was unique to Coeliac samples) [66], with increased butyrate production as found in the distal vessel of the Coeliac gut model after *C. pyrenoidosa* addition. Consistent with this, the inferred metagenomics analysis suggested reduced fermentation to acetate but increased conversion of acetyl-CoA into butyrate, particularly for the Coeliac donor. Other peculiar modifications in the functional profiles of the gut-derived microbial communities included, for both donors, a reduced potential for tryptophan biosynthesis, together with the increased contribution of pathways involved in nucleotides (i.e., guanosine and adenosine) degradation and biosynthesis. Taken together, these data with all the limitations of the study may indicate the propensity of *C. pyrenoidosa* to influence the fitness of the gut microbiota.

While Coeliac samples showed the greatest change in acetate concentration, significant increases in butyrate and propionate levels were observed for the healthy donor in vessels 2 and 3. The cell wall of *C. pyrenoidosa* is made up of various sugars, including rhamnose that can be used by *Ruminococcus* sp. to produce propionate via the propanediol pathway [67]. Interestingly, when rats were given *C. pyrenoidosa*, an increase in the relative abundance of *Ruminococcus* sp. was observed. This was negatively associated with fasting blood glucose, suggesting that the hypoglycaemic effect of *C. pyrenoidosa* could be due to its role in modulating the microbiome. *Ruminococcus* sp. are mostly found in herbivores and omnivores as they predominantly degrade cellulose [68]. An increase in *Ruminococcus* sp. was only observed in healthy donor samples after feeding with *C. pyrenoidosa*. In the healthy donor, substantial changes were also observed in the Bacteroidetes phylum with a decrease in Bacteroides coupled with an increase in Prevotella sp. in all 3 vessels after intervention. This suggests that *C. pyrenoidosa* can selectively improve the growth of Prevotella over Bacteroides sp. in certain microbiomes. Prevotella sp. produce the SCFA precursor succinate, which can be converted to propionate through the succinate pathway [69].

*C. pyrenoidosa* contains several different polysaccharides, cellulose, polyphenols and amino acids, and therefore the change seen in bacterial composition and SCFA production could be due to an individual component or a combined synergistic effect [23,66]. As evidenced, the Coeliac microbiota differs in composition and metabolite production from a healthy donor and consequently transforms in different ways in response to *Chlorella* fermentation. The role and effect of SCFAs on various systems in the body have been gaining recognition. Butyrate acts as a colonocyte energy source as well as improving gut barrier function, reducing inflammation and creating tolerance to commensal bacteria in the gut [69,70]. Increased butyrate production has also been shown to improve β-cell function [19]. The increase in *Faecalibacterium* sp. and butyrate production observed in both gut models could be beneficial for the host if it can be reproduced in vivo. *Faecalibacterium* sp. are dominant butyrate producers and can be used as an indicator of a healthy microbiota due to their beneficial effects on gut health, including anti-inflammatory and immunomodulatory properties [71,72], and insulin sensitivity [73]. As people with Coeliac disease usually have a lower abundance of *F. prausnitzii*, an increase in its proportions could improve gut inflammation and metabolic markers like insulin sensitivity. In all samples, the increase in *Faecalibacterium* sp. was observed especially in vessels representing the proximal and transverse colon. Butyrate producers are indeed rarely found in the distal part of the colon where the pH increases to 6.5 [14], indicating that fermentable fibre was abundant in this in vitro model or potentially acetate was converted to butyrate by the microbiota [67]. A stable isotope study in healthy individuals determined that 24% of acetate is converted into butyrate by the microbiota and was independent of butyrate-producing capacity [74].

## 5. Conclusions

Here, we have reported that the Coeliac-related microbiota differs in composition, predicted function and metabolite production from a healthy donor and consequently responds in different ways to *C. pyrenoidosa* in in vitro fermentation models. Despite the use of faecal samples from only one volunteer per type, our findings have shown that *C. pyrenoidosa* may lead to potential improvements in the intestinal microbial ecosystem, some common and others unique to the donor state.

In vitro gut fermentation systems have numerous challenges and limitations, especially the lack of host components. Furthermore, the reproducibility and functional stability of the human gut microbiota in in vitro gut fermentation models is frequently challenged and the subject of much criticism. Nevertheless, these models provide an innovative technological platform on which the greatest advantages are exhibited by the virtually unlimited experimental capacity sincethe experimentation is not restricted by ethical concerns. *C. pyrenoidosa* may thus manipulate the gut microbial population and alter metabolic activity towards a configuration that might confer health benefits to the host but still further research is required to extend our understanding of human intestinal health and the impact of *C. pyrenoidosa*.The preliminary data presented in this paper demonstrate all the strengths and limitations of this study and lay the foundations for the design of an in vivo human intervention study that needs to confirm the promising trends herein observed.

## Figures and Tables

**Figure 1 molecules-26-02330-f001:**
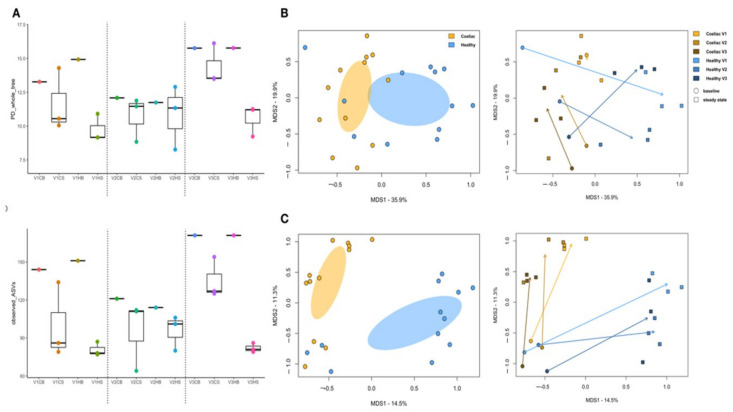
The faecal-derived microbial communities of the Coeliac donor (*n* = 1) differ from those of the healthy donor (*n* = 1) in fermentation experiments in the presence of *C. pyrenoidosa* extract (Sun Chlorella ‘A’ Powder). (**A**) Boxplots showing alpha diversity, measured according to Faith’s Phylogenetic Diversity index (PD whole tree) and observed ASV metrics, in fermentation samples from Coeliac (**C**) vs. healthy (H) donor, collected at baseline (SS1) (**B**) or after addition of *C. pyrenoidosa* upon reaching the steady state (SS2) (S), from the 3 vessels of the gut model (V1 vs. V2 vs. V3). Principal coordinates analysis (PCoA) based on weighted (**B**) and unweighted (**C**) UniFrac distances, showing all fermentation samples, stratified by sample origin (Coeliac vs. healthy, left panels) and fermentation condition/time point (vessel 1, 2 and 3; baseline vs. steady state; right panels). A significant separation between Coeliac and healthy samples was found regardless of all other variables (*p* ≤ 0.002, permutation test with pseudo-F ratio). The arrows indicate the direction of the temporal variations observed within the faecal-derived microbial communities of the Coeliac vs. healthy subject in the three vessels, from the baseline to the steady state.

**Figure 2 molecules-26-02330-f002:**
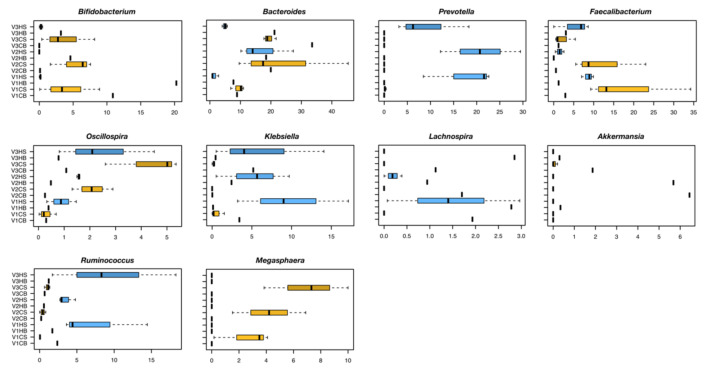
Genus-level differences in the faecal-derived microbial communities between the Coeliac (*n* = 1) and healthy donor (*n* = 1) following fermentation with *C. pyrenoidosa* extract (Sun Chlorella‘A’ Powder). Boxplots showing the relative abundance distribution of bacterial genera differentially represented among the study groups (vessels (V) 1, 2 and 3, inoculated with faeces from Coeliac (C) vs. healthy (H) donor, at baseline (B) and steady state (S) after addition of *C. pyrenoidosa* extract (Sun Chlorella ‘A’ Powder). (*p* ≤ 0.1, Kruskal−Wallis test).

**Figure 3 molecules-26-02330-f003:**
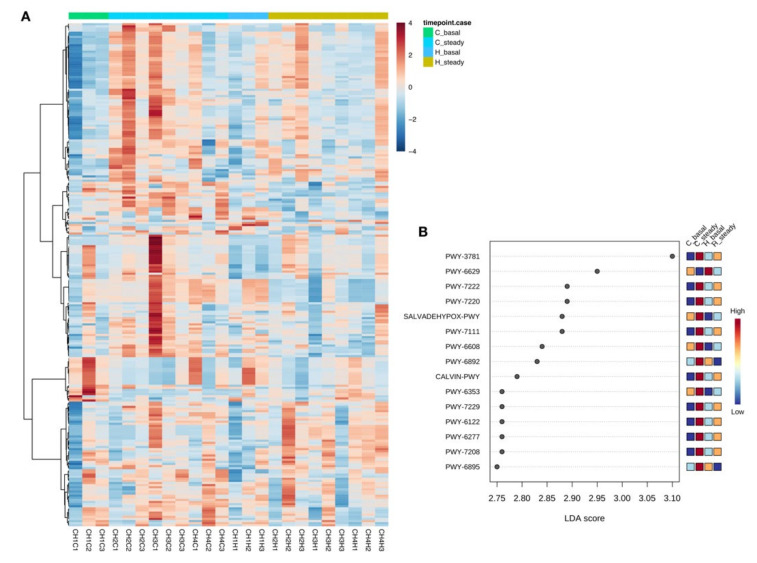
Functional differences in the faecal-derived microbial communities between Coeliac (*n* = 1) and healthy donor (*n* = 1) samples at baseline and following fermentation with *C. pyrenoidosa* extract (Sun Chlorella ‘A’ Powder).(**A**) Hierarchical clustering of the inferred metagenomes of the study groups (Coeliac (C) vs. healthy (H) donor, at baseline (basal) and steady state (steady) after addition of *C. pyrenoidosa* extract (Sun Chlorella ‘A’ Powder).The heatmap shows Ward-linkage clustering based on the Kendall’s correlation coefficients of the abundance profile of the 238 most abundant pathways identified. (**B**) Linear discriminant analysis effect size (LEfSe) analysis showing the differentially represented pathways upon addition of *C. pyrenoidosa* extract (Sun Chlorella ‘A’ Powder) for both C and H samples. The pathways are indicated with the MetaCyc ID.

**Figure 4 molecules-26-02330-f004:**
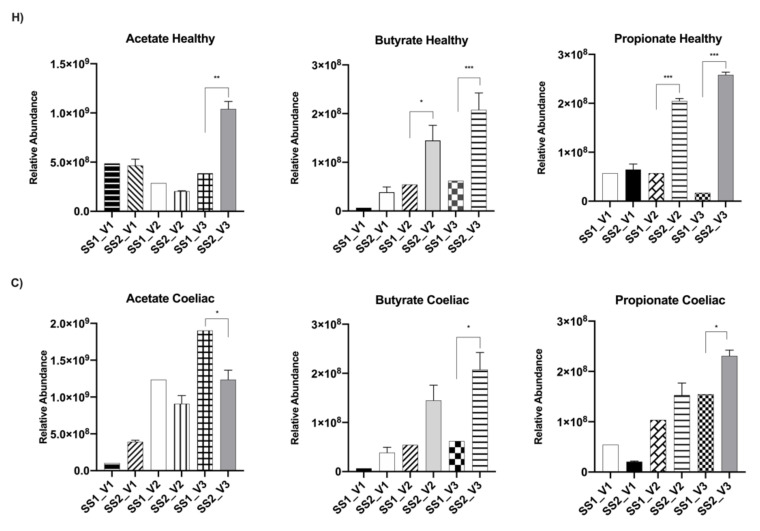
The faecal-derived microbial SCFAs (acetate, butyrate, propionate) of the healthy (*n* = 1) and Coeliac (*n* = 1) donors in fermentation experiments in the presence of *C. pyrenoidosa* extract (Sun Chlorella ‘A’ Powder). (**H**) ^1^H-NMR analysis of SCFAs relative abundance in vessels (V1, V2, V3) from healthy donor before (SS1) and following fermentation with *C. pyrenoidosa* extract (Sun Chlorella ‘A’ Powder). (SS2) (3 g/day). Significance of one-way ANOVA with Tukey’s posthoc comparison after treatment: * *p* < 0.05; ** *p* < 0.001 and *** *p* < 0.0001. (**C**) ^1^H-NMR analysis of SCFAs relative abundance in vessels (V1, V2, V3) from Coeliac donor before (SS1) and following fermentation with *C. pyrenoidosa* extract (Sun Chlorella ‘A’ Powder). (SS2) (3 g/day). Significance of one-way ANOVA with Tukey’s posthoc comparison after treatment: * *p* < 0.05; ** *p* < 0.001 and *** *p* < 0.0001.

## Data Availability

The datasets sequencing reads were deposited in the National Center for Biotechnology Information Sequence Read Archive (Bioproject PRJNA685154).

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
