# Peer review of "An In Vitro Pilot Fermentation Study on the Impact of Chlorella pyrenoidosa on Gut Microbiome Composition and Metabolites in Healthy and Coeliac Subjects"

_molecules, 2021, doi:10.3390/molecules26082330_

Round 1
Reviewer 1 Report
The manuscript describes a fermentation study of C. pyrenoidosa on gut microbiome in Healthy people and Coeliac patients. It is a well written paper.
A minor suggestion is that the authors should consider citing the following article:
Supplementation with Chlorella vulgaris, Chlorella protothecoides, and Schizochytrium sp. increases propionate-producing bacteria in in vitro human gut fermentation
Author Response
We thank the reviewer 1 for this comment and this reference highlighting the potential use of
Chlorella as a functional food has been now added and listed as new reference 25.
25. Jin, J.B.; Cha, J.W.; Shin, I.; Jeon, J.Y.; Cha, K.H.; Pan, C. Supplementation with
Chlorella vulgaris, Chlorella protothecoides, and Schizochytrium sp. increases
propionate‐producing bacteria in in vitro human gut fermentation. J Sci Food Agric 2020, 100, 2938-2945.
Reviewer 2 Report
General comments:
By using an in vitro simulated human colon model system, authors examined effects of green algae Chlorella pyrenoidosa on microbial profiles, which were generated from faecal samples collected from a healthy donor and a Coeliac donor. The study is interesting and the model system may have many potential applications. However, major drawbacks of the study are that only 2 biological samples were examined and microbial fingerprints were not generated directly from the faecal samples, which would provide a baseline for evaluation of microbial coverage by this model system.
Specific comments:
Page 6: The faecal samples should be directly used for DNA isolation and 16S gene sequencing to establish an original baseline. This should be done on the diluted and homogenized faecal samples as well to establish the second baseline.
Page 6: It is only mentioned that “samples (4.5 mL) were collected from all vessels of the colonic system and stored at -80 °C for future analysis”. Detailed information of sample collection should be provided. How many samples were collected from each vessel in SS1 and SS2 stage for DNA sequencing and for fatty acid analysis?
Page 9-12: The sample size (n=?) for data in all the figures should be written in the figure legend.
Author Response
Page 6: The faecal samples should be directly used for DNA isolation and 16S gene sequencing to establish an original baseline. This should be done on the diluted and homogenized faecal samples as well to establish the second baseline.
We thank the Reviewer for this comment. This point has been already addressed and validated in different in vitro fermentation papers. Changes in bacterial community dynamics have been observed in human gut models by the gut model medium composition and the model retention time. In the human gut model systems, the medium flowing through the model means that different nutrient availability occurs in different regions, enabling different environmental conditions to be modeled, supporting the growth and the establishment of different microbial populations at different enteric sites. The bacterial community within the human model system could be treated with different foods, bacteria, or antibiotics to determine the impact of dietary changes on the microbiota; this could therefore aid in determining whether certain changes may be of benefit or not to a human. Indeed, the advantage of a model system is that the diet is very well controlled; therefore, if a test substrate is added after steady state has been reached, subsequent changes will be known to be a result of the substrate. This is the reason to compare the microbiota of the SS1 and SS2. Even if the original baseline is not yet established at the time of the initial inoculum, it will reach the stable condition after the 8 turnover (SS1) via the feeding with the complex gut model medium. We have added now new references to support this point as follow:
- Macfarlane, G.T.; Macfarlane, S.; Gibson, G.R. Validation of a three-stage compound continuous culture system for investigating the effect of retention time on the ecology and metabolism of bacteria in the human colon. MicrobEcol 1998, 35, 180-187.
- Nissen, L.; Casciano, F.; Gianotti, A. Intestinal fermentation in vitro models to study food-induced gut microbiota shift: an updated review. FEMS Microbiol Lett 2020, 367, fnaa097.
- Byrd, D.A.; Chen, J.; Vogtmann, E.; Hullings, A.; Song, S.J.; Amir, A.; Kibriya, M.G.; Ahsan, H.; Chen, Y.; Nelson, H. Reproducibility, stability, and accuracy of microbial profiles by fecal sample collection method in three distinct populations. PloS one 2019, 14, e0224757.
- Rycroft, C.E.; Jones, M.R.; Gibson, G.R.; Rastall, R.A. A comparative in vitro evaluation of the fermentation properties of prebiotic oligosaccharides. J ApplMicrobiol 2001, 91, 878- 887.
- Leng, J.; Walton, G.; Swann, J.; Darby, A.; La Ragione, R.; Proudman, C. “Bowel on the Bench”: Proof of Concept of a Three-Stage, In Vitro Fermentation Model of the Equine Large Intestine. Appl Environ Microbiol 2019,
Page 6: It is only mentioned that “samples (4.5 mL) were collected from all vessels of the colonic system and stored at -80 °C for future analysis”. Detailed information of sample collection should be provided. How many samples were collected from each vessel in SS1 and SS2 stage for DNA sequencing and for fatty acid analysis?
We thank the reviewer for this point and it has been addressed accordingly. “A sample of 4.5 mL was taken through the sample port of each vessel (V1, V2 and V3) subsequent the first (SS1) and second full turnover (SS2) of medium through the model at 15, 16 and 17 days and at 33, 34 and 35 days, respectively. Two aliquots of 2.25 ml each were centrifuged at 13000 rpm, the supernatant was stored at −20°C for short chain fatty acid profilewhereas the pellets were stored at −80°C for 16S rRNA gene-based next-generation sequencing analysis”.
Page 9-12: The sample size (n=?) for data in all the figures should be written in the figure legend.
Apologies for missing this, the sample size (n) has been added accordingly in the figure legends.
Reviewer 3 Report
In the manuscript molecules-1128557, entitled “An in vitro Pilot Fermentation Study on the Impact of Chlorella pyrenoidosa on Gut Microbiome Composition and Metabolites in Healthy and Coeliac Subjects” by van der Linde et al., the authors present a pilot study that evaluates the consequences of C. pyrenoidosa algae on celiac and healthy microbiota, in terms of microbial composition and metabolic activities. For that purpose, they employ an in vitro system that mimics the human colon.
Although potentially interesting, since C. pyrenoidosa algae has an interest as a food supplement, and might have the potential to modulate gut microbiota composition and function, the study has clear limitations with regard to study sample and experimental design. The first limitation is that it only has 2 individuals (a woman suffering from celiac disease, and a healthy woman), making it impossible to make conclusions and extrapolate the results to the general celiac vs healthy population. The second limitation is that each of those individuals have followed a different diet: the celiac woman has had a GFD and the healthy woman has followed a normal gluten-containing diet, adding even more experimental variability to the study design. With this limitations in mind, I cannot recommend the publication if this manuscript in its present form.
However, if the authors want to publish an improved form of their study in the future, I would suggest the following major and minor changes:
Major points:
- The structure of the Introduction section is not clear (it is written in a single paragraph) which makes it difficult to get the main ideas, and understand the message that the authors want to give.
- The authors speak about Chlorella pyrenoidosa supplementation, but it is not clear how it is administered when it is commercially sold. In tablets as well?
- I would suggest to move Table 1 to the supplementary material (indeed, it is referenced as Table S1 in the text). I have never before seen DNA/RNA content in a table describing the nutritional composition of a food supplement: is this relevant? What does % of NRV Show NRV only mean?
- It is very hard for a non-expert reader to imagine this system... would it be possible to include a scheme or picture?
- Did the authors some kind of purification to distinguish between bacterial 16S rRNA and human mitocondrial RNA?
Minor points:
- Why do authors include numbers in the keywords?
- Figures, especially Figure 1, are very small, and the low resolution makes it very difficult to read what is written.
- I found a few typos or formatting mistakes:
- in vivo and in vitro terms are not always italicized
- Chlorella pyrenoidosa is not always italicized
- Sometimes Coeliac disease is abbreviated as “CD” and sometimes as “C”
Author Response
Journal: Molecules
Section: Natural Products Chemistry
Special Issue: Bioactive Compounds on Health and Disease
Manuscript ID: molecules-1128557
Article Type:Article
Number of Pages: 17
Title: An in vitro Pilot Fermentation Study on the Impact of Chlorella pyrenoidosa on Gut Microbiome Composition and Metabolites in Healthy and Coeliac Subjects
Authors:Carmen van der Linde, Monica Barone, Silvia Turroni, PatriziaBrigidi, EnverKeleszade, Jonathan R. Swann, and Adele Costabile
Comments of the Reviewers and Editor are in black, responses are in blue and citations from and changes in the manuscript are in red.
Reviewer 3 - Review Report (Round 1)
Comments and Suggestions for Authors
In the manuscript molecules-1128557, entitled “An in vitro Pilot Fermentation Study on the Impact of Chlorella pyrenoidosa on Gut Microbiome Composition and Metabolites in Healthy and Coeliac Subjects” by van der Linde et al., the authors present a pilot study that evaluates the consequences of C. pyrenoidosa algae on celiac and healthy microbiota, in terms of microbial composition and metabolic activities. For that purpose, they employ an in vitro system that mimics the human colon.
Although potentially interesting, since C. pyrenoidosa algae has an interest as a food supplement, and might have the potential to modulate gut microbiota composition and function, the study has clear limitations with regard to study sample and experimental design. The first limitation is that it only has 2 individuals (a woman suffering from celiac disease, and a healthy woman), making it impossible to make conclusions and extrapolate the results to the general celiac vs healthy population. The second limitation is that each of those individuals have followed a different diet: the celiac woman has had a GFD and the healthy woman has followed a normal gluten-containing diet, adding even more experimental variability to the study design. With this limitations in mind, I cannot recommend the publication if this manuscript in its present form.
However, if the authors want to publish an improved form of their study in the future, I would suggest the following major and minor changes:
Major points:
The structure of the Introduction section is not clear (it is written in a single paragraph) which makes it difficult to get the main ideas, and understand the message that the authors want to give.
We thank the reviewer 3 for this comment, and for better understanding and improving the readability of the introduction, we have revised the structure accordingly.
The authors speak about Chlorella pyrenoidosa supplementation, but it is not clear how it is administered when it is commercially sold. In tablets as well?
We apologies for this mistake andwe can confirm that for this study Sun Chlorella® 'A' Powder was used. This form is the only Chlorella that is pulverized by DYNO®-Mill. This remarkable technology liberates the nutrients in Chlorella, whilst breaking up the indigestible Chlorella cell wall without the need for excessive heat or chemicals. Now this information has added in the Materials and Methods to clarify how we have administrated this formulation.
I would suggest to move Table 1 to the supplementary material (indeed, it is referenced as Table S1 in the text). I have never before seen DNA/RNA content in a table describing the nutritional composition of a food supplement: is this relevant? What does % of NRV Show NRV only mean?
We thank the Reviewer for this comment and this Table was added as Supplementary Table. We apologies if we have sound unclear and for the typo errors. The CODEX NRVs(Nutrient reference values) are calculated based on recommended values (recommended dietary allowances-RDA and individual nutrient level for 98% of the population). Here we are presenting the % of NRV = Nutrient Reference Value and we are not showing the % of RDA according to Regulation (EU) 1169/2011. We have now revised and changed this Table in order to not cause any misleading or doubt.
It is very hard for a non-expert reader to imagine this system... would it be possible to include a scheme or picture?
We thank the Reviewer for this comment. We have now included a supplementary Figure (Figure S1) reporting the diagrammatic chart of the validated three-stage continuous system.
Did the authors some kind of purification to distinguish between bacterial 16S rRNA and human mitocondrial RNA?
We thank the Reviewer for this very good comment. In this study, we focused more in depth on the 16S rRNA gene-based next-generation sequencing. Metagenome prediction of Greengenes-picked ASVs was performed with PICRUSt2), using MetaCyc as reference for pathway annotation. Initially we thought that the metagenome potential could have been inferred using the PICRUSt2 pipeline. Thus we used the LEfSe algorithm to identify the MetaCyc pathways affected by the addition of the C. pyrenoidosa extract. According to this analysis, the faecal-derived microbial communities from healthy and Coeliac donors showed distinct functional profiles that are differentially modulated by C. pyrenoidosa. In particular, we found alterations in the pathways related to fermentation to short-chain fatty acids, consistent with our metabolic findings as also shown in Figure 3. Unfortunately, we could not perform any additional purification analysis due the funding available.
Minor points:
Why do authors include numbers in the keywords?
Apologies for this typo error and now has been revised accordingly.
Figures, especially Figure 1, are very small, and the low resolution makes it very difficult to read what is written.
We thank the reviewer for this comment and we have now improved the resolution of the Figure 1.
I found a few typos or formatting mistakes:
o in vivo and in vitro terms are not always italicized
o Chlorella pyrenoidosa is not always italicized
o Sometimes Coeliac disease is abbreviated as “CD” and sometimes as “C”
We can now confirm all this typo and formatting errors have been revised and corrected accordingly.

Round 2
Reviewer 3 Report
In this second version of the manuscript, the authors have made the suggested major and minor changes, but have not addressed the major limitation of the manuscript with regard to study sample and experimental design (only 2 study individuals with different diet). This was the major concern I had when I first read the paper, and still think it is a strong limitation. Therefore, I cannot recommend it for publication.
Author Response
We thank the Reviewer 3 for these comments. In vitro fermentation models have been developed and validated for studying the relationships between gut microbiota and food components. These models are increasingly used as an alternative to in vivo assays not only for disclosure of physiological activities of food components in the human intestine, but also for development of novel health functional foods. The robustness of each model depends on the certainty that any observed effects on microbial composition and metabolic activity are due solely to the applied experimental treatment and not to gut microbial adaptation to the simulated gut environment. This presumes that a compositional stability of the in vitro gut ecosystem is in place prior to applying any experimental treatment and that this pseudo-steady state can reproducibly generate identical microbial communities which are the basis for comparison with other treatments. In this context to reduce the potential antagonistic effects from a pool of several donors, we have decided to use faecal inoculum from just one donor as the microbiota of one donor might sense and defense the alien microbiota of a different donor. We have now revise and re-write the conclusions and part of the discussion to highlight all these limitations and the sample size.